# COVID-19 infodemic on Facebook and containment measures in Italy, United Kingdom and New Zealand

**Gabriele Etta**[1]*, **Alessandro Galeazzi**[2], **Jamie Ray Hutchings**[3], **Connor Stirling James Smith**[3], **Mauro Conti**[4], **Walter Quattrociocchi**[1], **Giulio Valentino Dalla Riva**[3]

**1** Center of Data Science and Complexity for Society, Department of Computer Science, Sapienza Università di Roma, Roma, Italy, **2** Department of Environmental Sciences, Informatics and Statistics, Ca' Foscari University of Venice, Venice, Italy, **3** School of Mathematics and Statistics, University of Canterbury, Canterbury, New Zealand, **4** Department of Mathematics, University of Padova, Padova, Italy

* quattrociocchi@di.uniroma1.it

**Data Availability Statement:** Data cannot be shared publicly because the study mainly relies on facebook posts obtained from Crowdtangle which, as it states in https://help.crowdtangle.com/en/

## Abstract

The COVID-19 pandemic has been characterized by a social media "infodemic": an over-abundance of information whose authenticity may not always be guaranteed. With the potential to lead individuals to harmful decisions for the society, this infodemic represents a severe threat to information security, public health and democracy. In this paper, we assess the interplay between the infodemic and specific aspects of the pandemic, such as the number of cases, the strictness of containment measures, and the news media coverage. We perform a comparative study on three countries that employed different managements of the COVID-19 pandemic in 2020—namely Italy, the United Kingdom, and New Zealand. We first analyze the three countries from an epidemiological perspective to characterize the impact of the pandemic and the strictness of the restrictions adopted. Then, we collect a total of 6 million posts from Facebook to describe user news consumption behaviors with respect to the reliability of such posts. Finally, we quantify the relationship between the number of posts published in each of the three countries and the number of confirmed cases, the strictness of the restrictions adopted, and the online news media coverage about the pandemic. Our results show that posts referring to reliable sources are consistently predominant in the news circulation, and that users engage more with reliable posts rather than with posts referring to questionable sources. Furthermore, our modelling results suggest that factors related to the epidemiological and informational ecosystems can serve as proxies to assess the evolution of the infodemic.

## Introduction

The World Health Organization (WHO) declared COVID-19 as a pandemic on March 11, 2020 [1]. To contrast its spreading, governments introduced a series of countermeasures known as Non-Pharmaceutical-Interventions (NPIs) [2, 3]: the adoption of facemasks, quarantines, social distancing, and mobility limitations [4]. The pandemic also sparked a high news volume

articles/4558716-understanding-and-citing-crowdtangle-data, cannot be shared in CSV format. However, any researcher can require access to CrowdTangle upon request. Our Supporting information files contain all the contents to guide the interested reader in replicating our study with information about CrowdTangle access and data collection methodology.

**Funding:** Authors WQ, GE, AG received funding from the 100683 EPID Project "Global Health Security Academic Research Coalition" SCH-00001-3391, provided by UK/G7. The funders had no role in study design, data collection and analysis, decision to publish, or preparation of the manuscript.

**Competing interests:** The authors have declared that no competing interests exist.

on social media [5], further increased by the start of the vaccination campaigns [6], which highlighted the critical role played by information in health emergencies [7, 8]. The novel news circulation dynamic has been defined as "infodemic" by WHO, a term indicating "an overabundance of information—some accurate and some not—that occurs during an epidemic" [9].

The study on the effects related to infodemics has been carried on for more than a decade. Indeed, Prof. Gunther Eysenbach, in 2002, coined the term *infodemiology* [10] to indicate "the study of the determinants and distribution of health information and misinformation—which may be useful in guiding health professionals and patients to the quality health information on the Internet. Information epidemiology, or infodemiology, identifies areas where there is a knowledge translation gap between best evidence (what some experts know) and practice (what most people do), as well as markers for 'high-quality' information". This study area offered the opportunity to develop a vast amount of applications [11–15] for surveillance, assessment, and investigation, whose aims are recognized with the term "infoveillance" [16].

In the context of opinion dynamics on social media, recent works highlighted how users tend to endorse information adhering to their beliefs, ignoring opposite points of view [17, 18]. This behavior promoted the emergence of homogeneous groups (that is, clusters in the social media networks) with similar news consumption dynamics, called echo chambers [19–23], and the creation of online environments sensitive to the diffusion of misinformation [24]. Further results pointed out that the formation and the segregation of such groups may be attributed to the role of platforms' feed algorithms that increase the distance among ideologically opposite groups of users [25, 26]. Further efforts aimed at quantifying the magnitude and the effects of infodemic on social media investigating a broader amount of platforms [5, 27, 28]. Authors from [29–31] focused on these effects by analyzing the topics, emotions, and concerns that emerged from such posts. Moreover, authors from [32] developed the Infodemic Risk Index (IRI) to quantify how users from each country can be susceptible to the spreading of unreliable news, demonstrating how the consumption of the latter decreases as the number of confirmed cases arose in the different countries.

In this paper, we compare the differential impact of the infodemic in three countries, considering both online and offline relevant factors: the focus of the news media coverage, the number of epidemic cases, and the strictness of the containment measures introduced by the governments (Stringency Index, SI). We analyse three countries—namely Italy, the United Kingdom, and New Zealand—that experienced a different time evolution of the pandemic and that implemented different containment strategies (see Section for further details). We first provide an epidemiological assessment of each country to characterize the impact of the pandemic and the strictness of the restrictions adopted. Then, we collect more than 6 million pieces of content talking about COVID-19 topics posted on Facebook in 2020 from public groups and pages. We categorize such posts into Questionable or Reliable, based on the trustworthiness of the referenced news outlets, to analyze the evolution and the engagement that both categories produced. Finally, we perform a regression on the number of posts by employing the number of COVID-19 confirmed cases, the strictness of the restriction policies adopted and the online news media coverage about the pandemic. Our results show that reliable posts are prevalent in the news circulated in 2020, across the three countries, and that users engaged more with such posts than with questionable posts in Italy and New Zealand (but not in UK) throughout the analysis period. Moreover, regression results suggest that the number of cases, the Stringency Index and the online news media coverage can serve as proxies to assess the evolution of the infodemic.

We provide a description of the country selection, the data collection process, the labeling of the posts concerning their news outlets and how we defined the metrics employed in this work.

## Materials and methods

### Country selection

We perform a comparative analysis on Italy, New Zealand, and the United Kingdom. The choice of these countries relies on the different strategies they employed in the mitigation of the virus during 2020. Indeed, Italy formerly implemented a series of containment measures only for those municipalities which were identified as the central spreaders of the virus [33]. Then, such measures were extended to regions following the same rationale, introducing the closure of schools and universities. The severity of the epidemiological profile led the country to establish a first national lockdown from 09/03/2020 to 18/05/2020, followed by a second one from 24/12/2020 to 06/01/2021 [34].

New Zealand, instead, adopted a series of countermeasures with the precise goal of eradicating the virus, characterized by early, rapid, and decisive government actions at various levels such as strict lockdowns, border-control, community, and case-based control measures [35]. Moreover, the country also enforced a nationwide lockdown that lasted from 23/03/2020 to 13/05/2020 [34], managing the spreading of the virus through a series of public health and social measures organized in a four-level COVID-19 Alert System [36].

Contrary to the first two countries, the United Kingdom opted for milder management of the pandemic in the early stages [37]. The rapid evolution of the pandemic, however, caused an immediate change of direction [38]. Indeed, the United Kingdom announced its intention to establish a lockdown, delegating its organization to its four constituent countries: England, Scotland, Wales, and Northern Ireland. Wales registered the most prolonged first lockdown from 23/03/2020 to 13/07/2020. Successively, the country enforced two more lockdowns in 2020 that lasted from 23/10/2020 to 11/12/2020 and 26/12/2020 to 02/04/2021 [34].

### Data collection

**Our World in Data.** The data concerning the evolution of the number of COVID-19 confirmed cases and deaths are obtained from Coronavirus Pandemic section of Our World in Data [39]. The website offers a worldwide set of statistics and metrics related to the COVID-19 pandemic, interactive visualizations, and data sources. In the context of this study, we retrieved the complete dataset that includes the daily statistics for all countries in the world since January 22, 2020. We retained only records from Italy, the United Kingdom, and New Zealand that referred to 2020.

**Facebook data.** The collection of posts on Facebook performed on a period that goes from 1/1/2020 to 31/12/2020. To provide a meaningful observation of the debated around COVID-19 topic, we created a list of keywords that included the general discussion around the COVID-19 outbreak and vaccination campaigns. Such dataset consists of a set of English lemmatized terms which were translated into Italian for consistency purposes (Refer to the complete list of terms used in Table 1).

The obtainment of the data was technically made through the employment of CrowdTangle [40], a Facebook-owned tool that tracks interactions on public content from Facebook pages,

**Table 1. List of terms employed in the data collection of posts on Facebook.** The asterisk represents a general string appearing before or after the term, depending on its position.

| Country | Terms |
| --- | --- |
| IT | covid*, *vax*, dose*, farma*, immun*, vacc*, sars*, lockdown, epidemia, pandemia, astrazeneca*, pfizer |
| UK & NZ | covid*, *vax*, dose*, pharma*, immun*, vacc*, sars*, lockdown, epidemic, pandemic, astrazeneca*, pfizer |

groups, and verified profiles. CrowdTangle does not include paid ads unless those ads began as organic, non-paid posts that were subsequently "boosted" using Facebook's advertising tools. CrowdTangle also does not include activity on private accounts or posts made visible only to specific groups of followers.

From a technical perspective, CrowdTangle does not provide a searching mechanism to retrieve all posts partially containing a provided input term. Therefore, we expanded the words listed in Table 1 to all their possible endings. The collection of posts from CrowdTangle included all pages that contained the search terms for the countries of interest.

## Characterizing reliability of posts by news outlet leaning

**Outlet classification.** To assess the trustworthiness of posts circulating on Facebook, we built a dataset of news outlets' domains from our original Facebook dataset where each domain is labeled either as *Questionable* or *Reliable*. The classification relied on a combination between Media Bias/Fact Check (MBFC) [41], an independent fact-checking organization, and a list of news outlets labelled as described in [42]. On MBFC, each news outlet is associated with a label that refers to its political bias, namely: *Right, Right-Center, Least-Biased, Left-Center, and Left*. Similarly, the website also provides a second label that expresses its reliability, categorizing outlets as *Conspiracy-Pseudoscience, Pro-Science* or *Questionable*. Noticeably, the *Questionable* set includes a wide range of political biases, from *Extreme Left* to *Extreme Right*. For instance, the *Right* label is associated with Fox News, the *Questionable* label to Breitbart (a famous right extremist outlet), and the *Pro-Science* label to *Science*. MBFC also provides a classification based on a *ranking bias score* that depends on four categories: *Biased Wording/Headlines, Factual/Sourcing, Story Choices*, and *Political Affiliation*. Each category is rated on a 0–10 scale, with 0 indicating the absence of bias and 10 indicating the presence of maximum bias. The *bias outlet score* is computed as the average of the four score categories. A different characterization is provided for humor and platforms websites, not accounting for the categorization process.

**Data collection and classification results.** The initial collection of posts from pages presented in Section produced a total of $\sim 6M$ posts from 218280 unique users. Then, we employed the classification process described in Section, obtaining 28631 categorized posts from 1785 unique users for New Zealand, 115421 posts from 14176 unique users for Italy and 1648283 posts Facebook from 111515 unique users for the United Kingdom. The result of the data collection process is described in Table 2.

## Metrics

**Stringency Index (SI).** To quantify the strictness of containment measures enforced by governments, we employ a measure called Stringency Index (SI) [43]. It is composed of nine indicators concerning containment and closure policies and health and system policies. The SI

**Table 2. Data breakdown of Facebook posts collected from pages whose admin was related to one of the three countries in exam, together with the results of the classification process.**

| Country | Total Posts | Categorized Posts with a Link | Questionable Posts | Reliable Posts |
|---------|-------------|-------------------------------|--------------------|----------------|
| NZ | 349 512 | 28 631 | 330 | 28 301 |
| UK | 3 382 628 | 520 520 | 51 465 | 469 055 |
| IT | 2 277 020 | 115 421 | 2 952 | 112 469 |

**Table 3. List of indicators included in the computation of Stringency Index (SI).** Each value encodes to a specific set of measures applied concerning the indicators. The Flag column indicates whether the policy has a variable that defines the geography of its application (1) or not (0).

| Indicator | Description | Max value (Nk) | Flag? (Fk) |
|---|---|---|---|
| School closing (C1) | Record closings of schools and universities | 3 | Yes = 1 |
| Workplace closing (C2) | Record closings of workplaces | 3 | Yes = 1 |
| Cancel public events (C3) | Record cancelling public events | 2 | Yes = 1 |
| Restrictions on gatherings (C4) | Record limits on gatherings | 4 | Yes = 1 |
| Close public transport (C5) | Record closing of public transport | 2 | Yes = 1 |
| Stay at home requirements (C6) | Record orders to "shelter-in-place" and otherwise confine to the home | 3 | Yes = 1 |
| Restrictions on internal movement (C7) | Record restrictions on internal movement between cities/regions | 2 | Yes = 1 |
| International travel controls (C8) | Record restrictions on international travel | 4 | No = 0 |
| Public information campaigns (H1) | Record presence of public info campaigns | 2 | Yes = 1 |

on a given day $t$ is computed through the following equation:

$$SI_t = \frac{1}{9} I_{j,t} \quad . \tag{1}$$

Each sub-index score $I$ for any of the nine indicators $j$ on any given day $t$ is computed as

$$I_{j,t} = 100 \frac{v_{j,t} - 0.5(F_j - f_{j,t})}{N_j}, \tag{2}$$

where $N_j$ is the maximum value of the indicator, $F_j$ specify whether the indicator has a flag variable ($F_j = 1$) or not ($F_j = 0$), $f_{j,t}$ the recorded binary flag for the indicator and $v_{j,t}$ the recorded policy value on the ordinal scale. A description of the indicators included in the computation of SI can be found in Table 3, while a complete description about the computation of the Stringency Index can be found at the following link [44].

**Weekly evolution of the engagement.** To assess users' engagement by outlet category, we used all possible types of user-post interaction –i.e., any type of reaction, share or comment. We then calculated the average weekly engagement to both the questionable and reliable sources as follows. Let $Q_w \subseteq Q$ and $R_w \subseteq R$ represent all the questionable and reliable posts that appeared in week $w$. Similarly, let $r_{Q_w}$, $c_{Q_w}$ and $s_{Q_w}$ represent the number of reactions, comments and shares for each questionable post at week $w$, whilst $r_{R_w}$, $c_{R_w}$ and $s_{R_w}$ refer to the number of reactions, comments and shares respectively for each reliable post at week $w$. Therefore, the average weekly engagement for questionable posts $\bar{e}_{q_w}$ at week $w$ can be defined as

$$\bar{e}_{q_w} = \frac{r_{Q_w} + c_{Q_w} + s_{Q_w}}{|Q_w|}, \tag{3}$$

whilst its counterpart $\bar{e}_{r_w}$ is defined as

$$\bar{e}_{r_w} = \frac{r_{R_w} + c_{R_w} + s_{Q_w}}{|R_w|}. \tag{4}$$

**GDELT Online News Coverage Index.** The GDELT (Global Database of Events, Language, and Tone) Project, powered by Google Jigsaw, is a database of global human society which "monitors the world's broadcast, print, and web news from nearly every corner of every country in over 100 languages and identifies the people, locations, organizations, themes, sources, emotions, counts, quotes, images and events driving our global society every second

of every day" [45]. Among the different services that GDELT provides, it supplies the translation of all the news they incorporate in 65 languages representing 98.4% of its daily non-English monitoring volume [46]. This feature allows to monitor the coverage that news media perform around specific topics. To this extent, we define as Online News Coverage Index (ONCI) a normalized measure of the percent of all global news coverage monitored by GDELT. It allows tracing how attention to a specific topic defined by the search keywords has changed over time and whether it is increasing or decreasing. In the current study, we made use of the public API provided by GDELT Online News Summary [47]. We set up the research with the following parameters:

- **Keyword(s)** field was set up with the following query: *"covid OR corona OR coronavirus OR covid-19 OR sars-cov-2 OR pfizer OR astrazeneca OR epidemic OR pandemic OR vaccine OR vaccins OR vaccination OR vaccinations OR lockdown"*

- **Time Period** fields were set up in order to be consistent with the one used in our analysis

- **Limit to Country** field contained the value of *United Kingdom*, *New Zealand* and *Italy*

- **Limit to Language** field was set up to *English* for New Zealand and the United Kingdom, *Italian* for Italy.

Since the ONCI depends on the coverage pursued by news media outlets from a specific country, the trends are not meant to be compared employing their absolute values. Instead, these trends can be compared by focusing on the changes that happened to them at a specific time.

## Results and discussion

### Evolution of COVID-19 outbreak

We first assess the epidemiological scenario for Italy (IT), New Zealand (NZ), and the United Kingdom (UK) during 2020. Fig 1 represents the cumulative number of confirmed cases, deaths and the average weekly evolution of the Stringency Index (SI). The SI is a scalar value

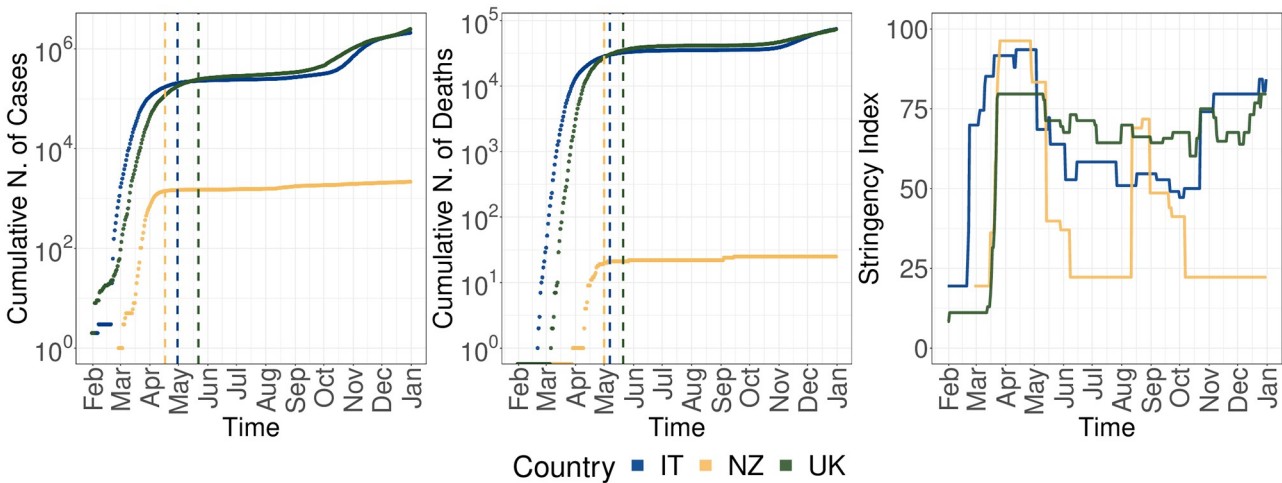

**Fig 1.** Overview of COVID-19 outbreak in 2020 depicted by: the cumulative number of confirmed cases (left) and deaths (middle), and the evolution of the Stringency Index (right) for Italy (IT—represented in blue), New Zealand (NZ—represented in yellow) and United Kingdom (UK—represented in green). Dashed lines represent the first plateau of the corresponding metric for each country, with a change ratio below than 1%.

that quantifies the strength of the containment measures adopted and it ranges from 0 (no effort adopted) to 100 (see Section in Materials and methods for more details).

From the left and middle panel of Fig 1 we observe how all countries were characterized by a rapid increase of both confirmed cases and deaths in the initial phase of the pandemic. New Zealand results in being the first country to reach a plateau in the evolution of the metrics, followed by Italy and the United Kingdom. In terms of containment measures, the right panel of Fig 1 shows how Italy is the country with the earliest response, with its SI value increasing from 19.44 to 69.91 on February 23rd. New Zealand results in being the country with the highest strictness measures, reaching a SI value of 96.30 on March 23rd in correspondence with the introduction of their first lockdown. Finally, the government of the United Kingdom, despite its delayed reaction, applied the restriction measures more permanently: it enforced 166 days of lockdown against the 77 from Italy and the 73 from New Zealand.

## Quantifying news consumption and online media coverage

To characterize the online flowing of information, we quantify the amount of questionable and reliable posts circulating in the three countries. To do so, we categorize posts into Questionable and Reliable according to the trustworthiness of the referred news source (see Section for more information). The left panel of Fig 2 shows how the news diet in the three countries is shaped by the presence of reliable posts. Moreover, right panel of Fig 2 describes how Italy

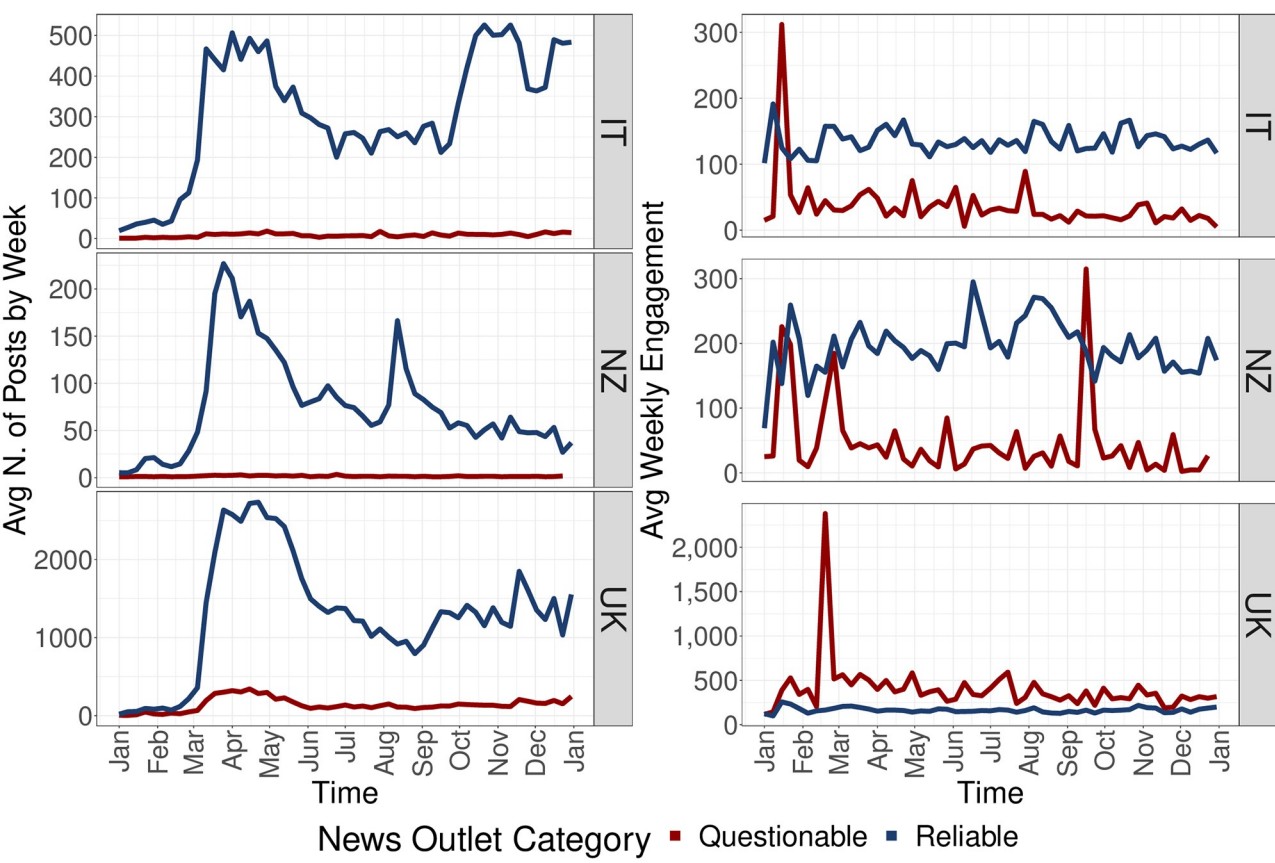

**Fig 2.** Left panel: Average weekly number of posts that circulated on Facebook for Italy (upper), New Zealand (middle) and the United Kingdom (bottom), with respect to the news outlet category. Right panel: Average weekly number of post interactions for Italy (upper), New Zealand (middle) and the United Kingdom (bottom), with respect to the news outlet category of the post the interactions refer to.

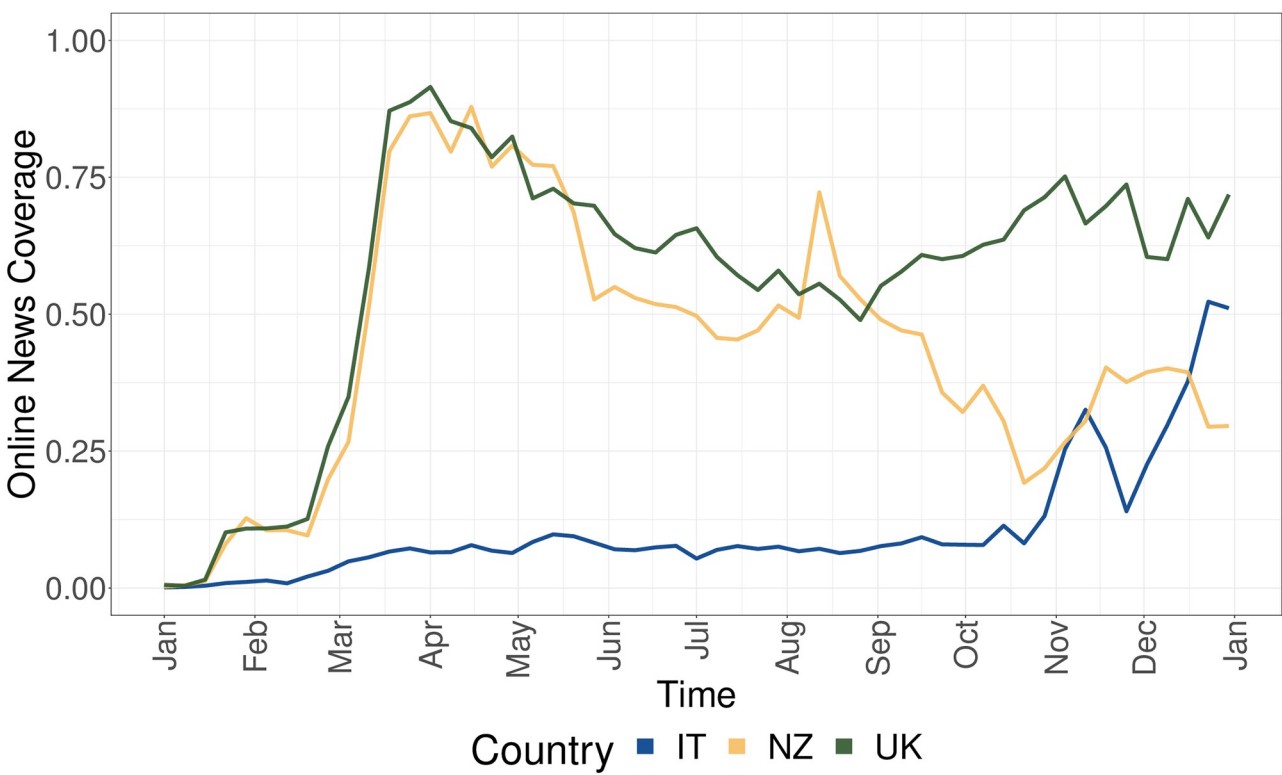

**Fig 3. Evolution of the normalized Online News Coverage Index (ONCI) for Italy (blue), New Zealand (yellow) and the United Kingdom (green).**

and New Zealand produce a higher level of engagement for reliable posts. The United Kingdom, oppositely, shows a preference towards questionable contents throughout all the year.

To further extend the study of the news consumption, we observe the evolution of the online news media coverage about COVID-19 in the three countries. We employ a measure called Online News Coverage Index (ONCI) computed on data from GDELT (Global Data on Events, Location, and Tone) Summary project (see Section for further details). Fig 3 shows the normalized evolution of the Online News Coverage Index (ONCI) for each of the three countries. We observe how coverage trends from New Zealand and United Kingdom resembles the posting behaviors in Fig 2, sustained by a Pearson score of 0.8 and 0.9 computed between ONCI and the evolution of the total number of posts in the two countries, respectively. Italy, instead, shows an online coverage trend which raises in the end of 2020, which may be attributable to the start of the vaccination campaigns.

In summary, we provide evidence about how the three countries are characterized by a reliable news diet about COVID-19 in 2020, with evolution in line with the online news media coverage about the COVID-19 topic. In terms of engagement, we observe that the United Kingdom is the only country where questionable posts receive more attention than reliable ones. In contrast, Italy and New Zealand are defined by a greater a engagement towards reliable news which is sustained throughout the entire analysis period.

## Assessing the role of COVID-19 factors in news consumption

To quantify the interplay between the evolution of the news circulating online and different aspects of the pandemic, we create a set of features from the ones presented in Section and.

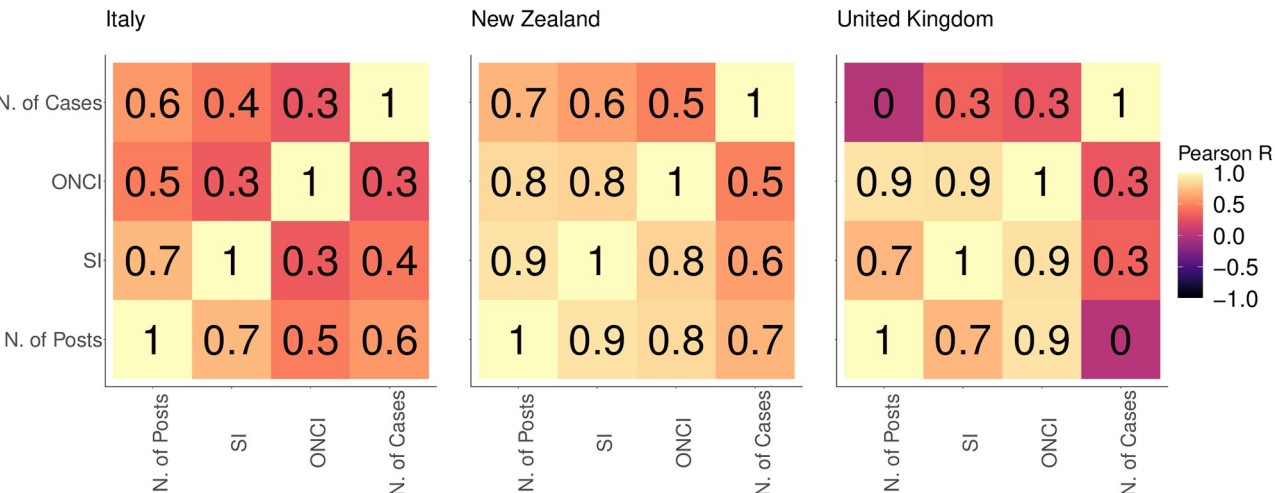

**Fig 4.** Pearson's correlation coefficient among ONCI (Online News Coverage Index), SI (Stringency Index), Number of COVID-19 Confirmed Cases and Posts computed on a weekly base for Italy (left), New Zealand (middle) and United Kingdom (right).

This set contains the SI, the confirmed number of COVID-19 cases and the ONCI. Fig 4 describes the relationship between such factors by means of Pearson's scores for Italy (left), New Zealand (middle) and the United Kingdom (right). We observe that SI has a strong positive correlation with the number of posts in all three countries with values that range from 0.7 to 0.9, meaning that the tightening of the restriction goes together with an increase of the online debate volume. ONCI, instead, has a correlation with the number of posts of 0.3 in Italy and 0.8 and 0.9 in the other countries, showing the consistency between the volumetric trend of the online news coverage in a country and the volume of the debate on social media. Finally, the number of cases shows a lack of correlation with the evolution of posts in United Kingdom, expressed by a Pearson score of 0.1, suggesting how the social media debate in the country may be related to further factors. Italy and New Zealand instead provide a stronger positive relationship with a value of 0.6 and 0.7 respectively, describing how the public debate is influenced by the severity of the pandemic outbreak.

For each of the three countries analyzed, we employ the factors considered in Fig 4 to model the evolution of the number of posts on Facebook by means of linear OLS regression of the form

$$N.ofPosts = \beta_0 + \beta_1 \text{ONCI} + \beta_2 \text{SI} + \beta_3 N. \, of \, Cases \qquad (5)$$

Results in Table 4 describe the estimators obtained from the regression in the three countries. In general, we observe that, for each of the three countries, the linear relationship between the *N. of Posts* and *ONCI* and *N. of Cases* is significant (at a level of 0.05). This suggests that the consumption of posts in social media is correlated to both the change in the epidemiological scenario and the pandemic media coverage. For Italy and New Zealand the *N. of Posts* published on Facebook is directly proportional to the three explanatory factors we employed in our model: in these countries, pages and groups are more likely to post when the epidemiological situation worsens, the pandemic news media coverage increases, and the mobility restriction measures becomes more strict. Instead, for the United Kingdom the correlation between the *N. of Posts* and the *N. of Cases* is negative (equal to −0.11 with a standard error of 0.02, significant at a level of 0.05). Moreover, the linear relation between the *N. of Posts* and the *SI* is not significant (at a level of 0.05), suggesting that the the strictness of the government response

**Table 4. Results for regression over the weeks with respect to the average number of posts in a country with multiple controls: ONCI, Stringency Index (SI) and the number of COVID-19 confirmed cases.**

| Country | Intercept | ONCI | bfSI | N. of Cases | $R^2$ |
|---|---|---|---|---|---|
| IT | -457.1 (1017) | 81510 * (35450) | 77.4 *** (15.08) | 0.078 ** (0.029) | 0.60 |
| UK | -3101 *** (849.4) | 8649 *** (854.4) | -51.20 . (26.06) | -0.11 *** (0.02) | 0.85 |
| NZ | -243.89 * (111.38) | 5617.18 *** (1383.44) | 12.82 *** (3.17) | 13.69 *** (3.57) | 0.87 |

The standard errors of the coefficients are reported in parenthesis, whilst the asterisks refer to the significance of their p-values in the following way:

*** $P < 0.001$,

** $P < 0.01$,

* $P < 0.05$,

. $P < 0.1$,

'' $P < 1$.

is not correlated with the trend of the infodemic. Thus, for the United Kingdom the infodemic volume increases with the news media coverage volume and decreases with the number of COVID-19 cases.

## Conclusions

In this work, we perform a comparative analysis on three countries, namely Italy, New Zealand, and the United Kingdom, which experienced different epidemic scenarios and adopted different containment strategies for the COVID-19 outbreak. First, we quantify the epidemiological dynamics and the strictness of the containment measures that characterized each country. Then, we estimate the potential volume of the infodemic with the number of posts circulating, the user engagement that the posts sparked, and coverage of COVID-19 by the national news media. Finally, we study the correlation between the spread of information and several factors expressing different facets of the pandemic and infodemic.

Our findings show that the infodemic evolution may increase with the volume of the online news coverage about COVID-19 in all the three countries. Moreover, the containment measures tightening are correlate with a potential increase in the volume of the infodemic in Italy and New Zealand, but its effect is not significant in UK. In terms of news diet composition, we observe a predominant circulation of reliable news. Indeed, in Italy and New Zealand users interact more with reliable sources, while in the United Kingdom they appear to be more involved with news referring to questionable sources.

Our regression results suggest that the infodemic may have a significant relationship with both socio-epidemiological factors, such as the number of cases and the strictness of the containment measures, and the attention of the traditional media, here quantified by the volume of the news media coverage. More specifically, the increase of pieces of contents in the United Kingdom appears to be associated mainly with the evolution of the online news media coverage, whilst any increase in the epidemiological metrics is associated with a reduction of posts circulating. In Italy and in New Zealand the growth of the infodemic is associated with an increase in any of the factors considered.

We find that the ongoing COVID-19 pandemic has a complex interaction with the evolution of the infodemic, both from a user and a media perspective. From a governance perspective, these findings raise the urgency to develop public health regulations that recognize the interplay between the online and the offline world. "Indeed, the existence of a role for the

number of cases, the news media coverage, and the strictness of containment measures in the evolution of infodemic is a pivotal point for policy-makers to take decisions for the safety of community members. Such decisions should consider that effective communication strategies must be developed and followed when publishing and circulating news, knowing that people, especially during a severe epidemic spreading, may be more susceptible to the content proposed. Furthermore, further actions are required to increase citizen awareness and understanding of the pandemic. A concrete approach may include the development of epidemic literacy programs, the introduction of interdisciplinary observatories that quantify the interconnection between the two ecosystems with the involvement of impacted communities.

A major limitation to the analysis we conducted is the availability of social networks data. First, as CrowdTangle does not provide the content of the comments, we cannot investigate whether people are inclined to spam the comment section with misleading claims even if the original post only presents credible information. We are aware that this methodology reduces the study sample, which CrowdTangle already restricts since it provides only posts from public Facebook pages with more than 25K Page Likes or Followers, public Facebook groups with at least 95K members, all US-based public groups with at least 2K members and all verified profiles. [48]. Furthermore, the prevalence of reliable news may evidence the positive commitment expressed by Facebook [49] to contrast the spreading of COVID-19 misinformation. Yet, this can not be verified lacking any transparent information about removed posts, groups, and pages. The importance of infodemic dynamics for our public health invites to plead for a more transparent access to social media news consumption data.

Future of this study may include the scaling of these results on a broader scale, extending the number of countries and social media analyzed, and localizing the study to explore the differential impact of the infodemic on communities more targeted by disinformation [50].

## Supporting information

**S1 File. Data collection and filtering procedure.**
(PDF)

## Author Contributions

**Conceptualization:** Gabriele Etta, Alessandro Galeazzi, Jamie Ray Hutchings, Connor Stirling James Smith, Mauro Conti, Walter Quattrociocchi, Giulio Valentino Dalla Riva.

**Data curation:** Gabriele Etta, Alessandro Galeazzi, Jamie Ray Hutchings, Connor Stirling James Smith, Giulio Valentino Dalla Riva.

**Formal analysis:** Gabriele Etta, Giulio Valentino Dalla Riva.

**Funding acquisition:** Mauro Conti, Walter Quattrociocchi.

**Investigation:** Gabriele Etta, Alessandro Galeazzi, Jamie Ray Hutchings, Walter Quattrociocchi, Giulio Valentino Dalla Riva.

**Methodology:** Gabriele Etta, Alessandro Galeazzi, Giulio Valentino Dalla Riva.

**Project administration:** Gabriele Etta, Walter Quattrociocchi, Giulio Valentino Dalla Riva.

**Resources:** Gabriele Etta, Giulio Valentino Dalla Riva.

**Software:** Gabriele Etta, Alessandro Galeazzi, Giulio Valentino Dalla Riva.

**Supervision:** Gabriele Etta, Alessandro Galeazzi, Walter Quattrociocchi, Giulio Valentino Dalla Riva.

**Validation:** Gabriele Etta, Alessandro Galeazzi, Giulio Valentino Dalla Riva.

**Visualization:** Gabriele Etta, Giulio Valentino Dalla Riva.

**Writing – original draft:** Gabriele Etta, Alessandro Galeazzi, Mauro Conti, Giulio Valentino Dalla Riva.

**Writing – review & editing:** Gabriele Etta, Alessandro Galeazzi, Mauro Conti, Walter Quattrociocchi, Giulio Valentino Dalla Riva.

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
