## [Decision Letter · Decision Letter 0]

1 Mar 2022

PONE-D-22-02799COVID-19 infodemic on Facebook and containment measures in Italy, United Kingdom and New ZealandPLOS ONE

Dear Dr. ETTA,

Thank you for submitting your manuscript to PLOS ONE. After careful consideration, we feel that it has merit but does not fully meet PLOS ONE’s publication criteria as it currently stands. Therefore, we invite you to submit a revised version of the manuscript that addresses the points raised during the review process.

We look forward to receiving your revised manuscript.

Kind regards,

Mohamed F. Jalloh, PhD, MPH

Academic Editor

PLOS ONE

Journal Requirements:

"The authors acknowledge the 100683EPID Project “Global Health Security Academic Research Coalition” SCH-00001-3391."

"Authors WQ, GE, AG received funding from the 100683 EPID Project "Global Health Security Academic Research Coalition'' SCH-00001-3391, provided by UK/G7. The funders had no role in study design, data collection and analysis, decision to publish, or preparation of the manuscript."

4. Please upload a copy of Figures 1, 2,3 and 4, to which you refer in your text on pages 6 and 7. If the figure is no longer to be included as part of the submission please remove all reference to it within the text.

5. Please include a copy of Tables 1, 2 and 3 which you refer to in your text on pages 4 and 8.

Reviewers' comments:

Reviewer's Responses to Questions

**Comments to the Author**

1. Is the manuscript technically sound, and do the data support the conclusions?

Reviewer #1: Yes

Reviewer #2: Partly

2. Has the statistical analysis been performed appropriately and rigorously? 

Reviewer #1: Yes

Reviewer #2: Yes

3. Have the authors made all data underlying the findings in their manuscript fully available?

Reviewer #1: Yes

Reviewer #2: No

4. Is the manuscript presented in an intelligible fashion and written in standard English?

Reviewer #1: Yes

Reviewer #2: Yes

5. Review Comments to the Author

Reviewer #1: Paper is clearly written and lays out methodology and findings clearly and concisely. The association between the infodemic and other factors (SI, case count) are clear but recommend including more detail in the discussion on how these interactions affect the health and health behaviors of community members.

Reviewer #2: This study uses a novel approach to examine the relationship between the information environment of a specific social media platform and broader COVID-19 epidemiological/social developments. The study findings have important implications for understanding the dynamics of infodemics in the context of other public health data, including epidemiological data.

However, there are some important methodological limitations/questions that warrant further discussion, which are listed below. Some of these limitations ideally should be addressed in the discussion section.

1.) The list of keywords used to identify Facebook posts appears to be narrow. For example, it does not have slangs or other indigenous categories likely to be used by disinformation actors or misinformed consumers. The dataset may therefore be inherently biased towards more "credible" posts.

2.) Binary categorization of "questionable" vs. "reliable" Facebook posts seems somewhat limiting. A post can have a link to a credible source and still interpret it in a misleading way, or people can spam the comment section with misleading claims even if the original post only presents credible information.

3.) The fact that the co-authors only focus on posts with a link significantly reduces the amount of data analyzed - this is a bit concerning especially because CrowdTangle is already limiting in the sense that it only picks up a subset of all posts available on Facebook i.e., public content from verified accounts.

4.) The MBFC scale used to classify posts appears to be based on the US political landscape according to their website. It's a bit unclear if this is the most appropriate system to categorize content from Italy, UK, NZ. Suggest explaining in more detail the rationale behind using this scale.

5.) Can engagement with Facebook posts also be assessed in the regression analysis? If the goal is to characterize the relationship between information consumption and epidemiological situation/pandemic media coverage, I would think that data about engagement should be included in the model as opposed to just looking at number of posts?

6. ) I suggest reconsidering making conclusive claims about "infodemic volume" or "trend of the infodemic" because this study only looks at one specific aspect of the infodemic - I don't think we can assume that data from Facebook and ONCI can be used as proxy measures for broader information environment, which includes other social media platforms, closed online groups, offline conversations, etc. (also, again, data from CrowdTangle is restricted to public content and excludes content generated by "regular" consumers").

6. PLOS authors have the option to publish the peer review history of their article (what does this mean?). If published, this will include your full peer review and any attached files.

Reviewer #1: No

Reviewer #2: **Yes: **Atsuyoshi Ishizumi

---

## [Author Response · Author response to Decision Letter 0]

30 Mar 2022

For the entire response please refer to Response.pdf

---

## [Editor Report · Decision Letter 1]

1 Apr 2022

COVID-19 infodemic on Facebook and containment measures in Italy, United Kingdom and New Zealand

PONE-D-22-02799R1

Dear Dr. ETTA,

We’re pleased to inform you that your manuscript has been judged scientifically suitable for publication and will be formally accepted for publication once it meets all outstanding technical requirements.

Kind regards,

Mohamed F. Jalloh, PhD, MPH

Academic Editor

PLOS ONE
---

## [Editor Report · Acceptance letter]

5 May 2022

PONE-D-22-02799R1 

COVID-19 infodemic on Facebook and containment measures in Italy, United Kingdom and New Zealand 

Dear Dr. ETTA:

I'm pleased to inform you that your manuscript has been deemed suitable for publication in PLOS ONE. Congratulations! Your manuscript is now with our production department. 

Kind regards, 

on behalf of

Dr. Mohamed F. Jalloh 

Academic Editor

PLOS ONE